# Optimization Design of Automotive Body Stiffness Using a Boundary Hybrid Genetic Algorithm

**Haolong Zhong [1], Ting Xu [1],\*, Jianglin Yang [1], Meng Sun [1] and Fei Gao [2]**

[1] Ji Hua Laboratory, Foshan 528200, China
[2] State Key Laboratory of Automotive Simulation and Control, Changchun 130025, China
\* Correspondence: xuting@jihualab.com

**Abstract:** At the conceptual design stage, it is critical to use appropriate structural analysis and optimization methods. The thin-walled beam transfer matrix method (TBTMM) is adopted to establish the mathematical model of the simplified vehicle body-in-white (BIW) structure in this paper and compare it with the results of the finite element method (S-FEM) to verify the approach. In addition, on the basis of the boundary simulation genetic algorithm (BSGA) and local search procedure, a boundary hybrid genetic algorithm (BHGA) is proposed. BHGA is benchmarked on 20 test functions and is compared with current meta-heuristic algorithms to prove its effectiveness and universality. Finally, considering the bending and torsional stiffness constraints, BIW conceptual model is lightweight and designed with an optimizer.

**Keywords:** thin-walled beam transfer matrix method; BIW structure; boundary hybrid genetic algorithm; lightweight design

## 1. Introduction

Compared with other stages, the conceptual design stage has a higher degree of design freedom and various requirements that are easier to meet, which is crucial for vehicle innovation, cost saving, and design cycle shortening [1]. The bending stiffness, torsional stiffness, and NVH performance of the body-in-white (BIW) structure are critical to the safety and comfortability of the vehicle, while the weight performance of the BIW has an effect on cost saving. There is a direct correlation between these two performances; that is, the reduction of vehicle weight requirement will lead to an overall reduction of stiffness and NVH performance. In order to obtain the optimal BIW structure that meets various requirements in the conceptual design, a lightweight design is selected to coordinate vehicle weight (objective function) and its performance (constraints). The performance of bending and torsional stiffness will be considered in this paper.

It is common to establish the mechanical model of the BIW by employing the finite element method (FEM). Bai et al. [2] described a simplified finite element model to provide early-stage predictions of a detailed model. A concept CAE modeling approach based on FE models was presented by Donders et al. [1] to analyze and optimize the structural behaviors of the vehicle BIW. In addition, Mundo et al. [3] proposed a similar approach for replacing beam structures and joints in vehicle BIW. However, few CAD data are available at the conceptual design stage. Meanwhile, most approaches to FEM require a time-consuming FE model and can not obtain clear mathematical relationships between beam section properties and structural performances. Therefore, Qin et al. [4] developed an object-oriented MATLAB toolbox based on the exact transfer stiffness matrix method to calculate the static and dynamic performances. Later, Liu et al. [5] developed a mathematical method based on the reverberation ray matrix method to promote the conceptual design. The vehicle BIW can be regarded as consisting of thin-walled beams, so the warping deformation (longitudinal displacement) in the longitudinal direction of the beam due to

torsion should be considered [6]. However, the aforementioned studies are all based on the traditional beam theory, according to which the warping of the beam is not considered. Zhong et al. [7] also pointed out that warping has a great impact on the performances of thin-walled frame structures and developed the thin-walled beam transfer matrix method (TBTMM). Therefore, the TBTMM is adopted in this paper to establish the relationship between the section properties and structural mechanical performances so as to improve the calculation accuracy.

In order to realize the lightweight design of vehicle body-in-white (BIW) at the conceptual design stage, a boundary hybrid genetic algorithm (BHGA) was proposed to solve the engineering-constrained optimization problems. In the next section, BIW mathematical models for bending and torsional stiffness will be established. In Sections 3 and 4, the development and validation of BHGA will be introduced. Finally, the lightweight design is achieved by using BHGA.

## 2. Formulation of Vehicle BIW Mathematical Model

### 2.1. Formulation of Cross-Sectional Properties

In this paper, a simplified BIW model with rectangular thin-walled beams is adopted to analyze the mechanical performances, as shown in Figure 1. Different from the cross-section properties of general beam theory (the inertia moments $I_y$ and $I_z$, the cross-sectional area $A$, and the torsional constants $J$), the thin-walled beam theory involves fifteen cross-sectional properties, i.e., the angle of roll $\phi$, the cross-sectional area $A$, the shear areas $A_{sy}$ and $A_{sz}$ in the $y$- and $z$-directions, the hybrid shear area of the cross-section $A_{syz}$, the sectorial static moments $S_{sy}$ and $S_{sz}$ about the $y$- and $z$-axes, the warping torsion moment of inertia $I_p$, the St.Venant torsional constant corresponding to Bredt's shear stress $I_B$, the St.Venant torsional constant corresponding to St.Venant's shear stress $I_s$, the inertia moments $I_y$ and $I_z$, the sectional moment of inertia $I_w$, and the coordinates of the shear center $y_s$ and $z_s$ relative to the centroid. Thus, the thin-walled beam cross-section is defined as in Figure 2, in which the right-hand orthogonal coordinate system and the s-coordinate system are adopted, the latter being along the midline of the cross-section. Moreover, the width (*b*), height (*a*), and thickness (*t*) are defined as design variables. The rectangular thin-walled beam is used in this paper; that is, the shear center *S* coincides with centroid *C* and $\phi$ is equal to zero. The calculation formula for section properties has been deduced in detail in the previous paper [7].

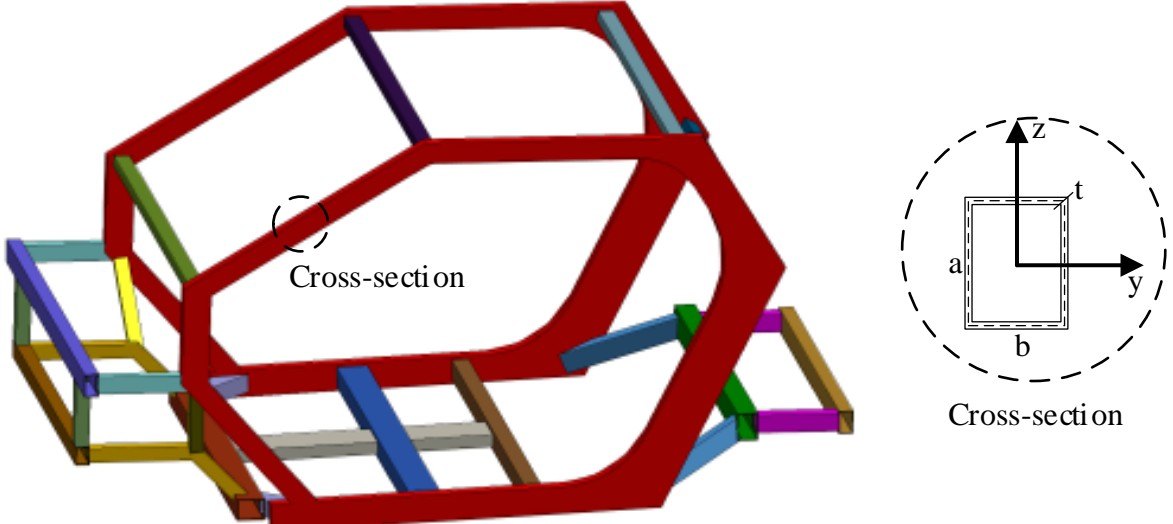

**Figure 1.** The conceptual BIW model.

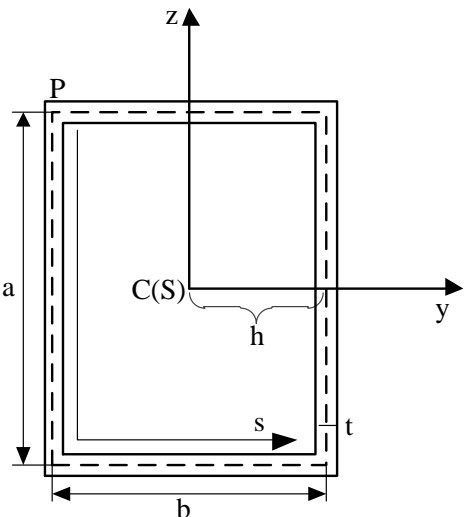

**Figure 2.** The details of the cross-section.

### 2.2. TBTMM Mathematical Model of BIW Structure

As shown in Figure 3, the mathematical model of the BIW conceptual structure consists of 53 beam members and 36 joints. The *i*th beam is identified by '①', and the *j*th joint is identified by '*j*'. According to thin-walled beam theory, the governing equations of beam members can be expressed as follows

$$\frac{d\mathbf{S}(x)_k}{dx} = \mathbf{H}_k\mathbf{S}(x)_k \tag{1}$$

where the $\mathbf{S}(x)_k$ is the state vector of kth member in the place of *x* containing seven displacement fields and seven force fields. $\mathbf{H}_k$ is expressed as the state function of *k*th member.

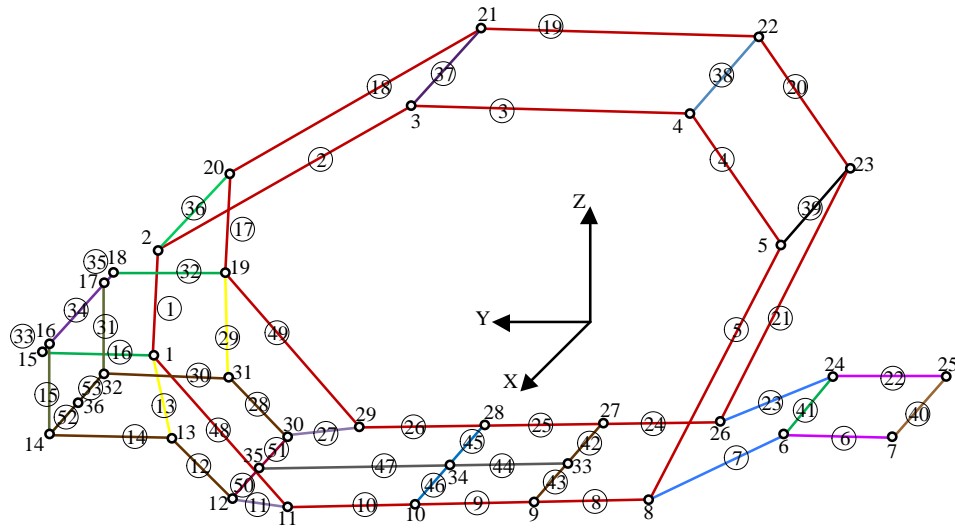

**Figure 3.** The mathematic simulation model of BIW conceptual structure.

The Laplace transform is used to solve Equation (1).

$$\mathbf{S}(x)_k = L^{-1}((X\mathbf{E} - \mathbf{H}_k)^{-1})\mathbf{S}(0)_k \tag{2}$$

where $L^{-1}()$ denotes the inverse Laplace transform and *X* is the Laplacian operator.

Then, the transfer matrix $\mathbf{T}_k$ can be obtained by substituting the length of beam $l_k$ into Equation (2), i.e., $\mathbf{T}_k = L^{-1}((X\mathbf{E} - \mathbf{H}_k)^{-1})\big|_{x=l_k}$, and the equation can be rewritten as Equation (3). The $\mathbf{T}_k$ is a constant matrix containing all fifteen cross-sectional properties.

$$\mathbf{S}(l_k)_k = \mathbf{T}_k \mathbf{S}(0)_k \tag{3}$$

When the transfer matrices of beams have been determined, the mathematical model for each joint can be assembled as follows

$$\overline{\mathbf{C}}_i \overline{\mathbf{P}}_i = \overline{\mathbf{f}}_i \tag{4}$$

where $\overline{\mathbf{C}}_i$ denotes the ith joint coupling matrix; $\overline{\mathbf{P}}_i$ contains all the state vectors defined at ith joint; $\overline{\mathbf{f}}_i$ is the matrix about external forces and moments at the ith joint. In general, the boundary conditions of the structures are homogenous; that is, each joint coupling matrix has half as many rows as columns. Moreover, each state vector contained in $\overline{\mathbf{P}}_i$ is one of the previously defined state vectors $\mathbf{S}(l_k)_k$ or $\mathbf{S}(0)_k$.

The global mathematical model of the BIW conceptual structure can be assembled as conventional FEM by combining joint coupling matrices.

$$\mathbf{CP} = \mathbf{f} \tag{5}$$

where $\mathbf{C}$ is a $742 \times 1484$ matrix, and $\mathbf{C} = diag[\overline{\mathbf{C}}_1, \overline{\mathbf{C}}_2, \ldots \overline{\mathbf{C}}_{36}]$ denotes the global joint coupling matrix; $\mathbf{P} = [\overline{\mathbf{P}}_1, \overline{\mathbf{P}}_2, \ldots \overline{\mathbf{P}}_{36}]^T$ contains all the state vectors, and is a $1484 \times 1$ matrix; $\mathbf{f} = [\overline{\mathbf{f}}_1, \overline{\mathbf{f}}_2, \ldots \overline{\mathbf{f}}_{36}]^T$ is the matrix about external forces and moments, and is a $742 \times 1$ matrix.

From Equations (3) and (5), $\mathbf{P}$ can be written as

$$\mathbf{P} = \mathbf{RT}_{global}\mathbf{Y} \tag{6}$$

where $\mathbf{Y}$ is a $742 \times 1$ matrix, and $\mathbf{Y} = [\mathbf{S}(0)_1, \mathbf{S}(0)_2, \cdots \mathbf{S}(0)_{53}]^T$ contains all the input state vectors defined at the left ends of the elements; $\mathbf{R}$ is a $1484 \times 1484$ matrix used to rearrange the order of the state vectors; $\mathbf{T}_{global}$ is the total transfer matrix and can be expressed as follows

$$\mathbf{T}_{global} = \begin{bmatrix} \mathbf{T}_1 & & & & \\ & \mathbf{T}_2 & & & \\ & & \ddots & & \\ & & & & \mathbf{T}_{53} \\ \mathbf{I}_1 & & & & \\ & \mathbf{I}_2 & & & \\ & & \ddots & & \\ & & & & \mathbf{I}_{53} \end{bmatrix} \tag{7}$$

where the transfer matrix $\mathbf{T}_i$ $(i = 1, 2, \ldots 53)$ is a $14 \times 14$ matrix and $\mathbf{I}_i$ $(i = 1, 2, \ldots 53)$ is an identity matrix, so the $\mathbf{T}_{global}$ is a $1484 \times 742$ matrix.

Then, the complete description of the BIW structure with respect to the state vectors $\mathbf{Y}$ and the external forces $\mathbf{f}$ can be rewritten as follows

$$\mathbf{CRT}_{global}\mathbf{Y} = \mathbf{f} \tag{8}$$

The $\mathbf{CRT}_{global}$ in Equation (8) is a $742 \times 742$ square matrix with respect to cross-sectional properties. Moreover, the external forces are set as the known values with respect to the boundary and load conditions. Then, Equation (8) may be formally solved by

$$\mathbf{Y} = [\mathbf{CRT}_{global}]^{-1}\mathbf{f} \tag{9}$$

### 2.3. Static Load Cases and Boundary Conditions

In this paper, the loads and boundary conditions of the BIW static analysis are shown in Figure 4. For the bending condition, the external forces $F_b$ are set as 1668 N along the Z- direction act on joints 10 and 28, respectively, and the constraint points are set on joints 6, 13, 24, and 31, respectively. The digital 1, 2, and 3 in the triangle region represent that the displacements in the global X-, Y-, and Z-directions are restrained to zero. For the torsion condition, the external forces $F_t$ are set as 1668 N act on joints 13 and 31, and the constraint points are set on joints 6, 24, and 36, respectively.

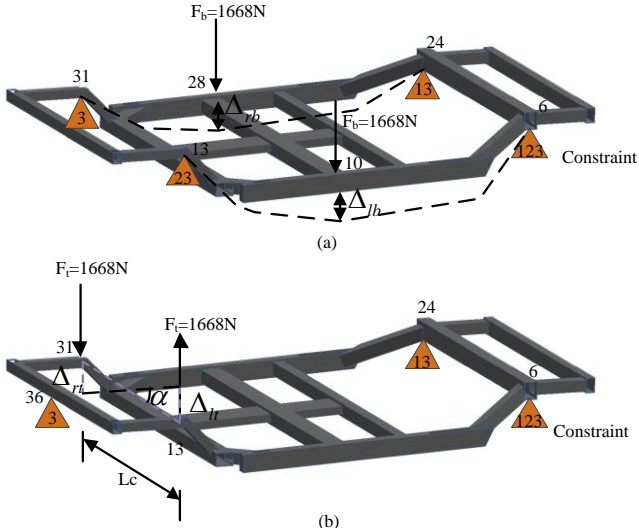

**Figure 4.** The static load cases and boundary conditions of BIW structure: (**a**) bending condition, (**b**) torsion condition.

Substituting the load cases and the boundary conditions into Equation (9) would yield the coupling equations of the whole structure. The maximum displacements of load directions $\Delta_{lb}$ ($\Delta_{rb}$) and $\Delta_{lt}$ ($\Delta_{rt}$) can be obtained by solving the coupling equations, respectively. And the bending stiffness $K_b$ and the torsion stiffness $K_t$ can be defined as follows:

$$\begin{cases} K_b = -\dfrac{4F_b}{\Delta_{lb}+\Delta_{rb}} \\ K_t = \dfrac{F_b Lc}{\alpha} \end{cases} \tag{10}$$

where $Lc$ is the distance between the two load points; $\alpha$ is the twist angle which can be calculated as follows:

$$\alpha = \frac{180(\Delta_{lt} - \Delta_{rt})}{\pi Lc} \tag{11}$$

### 2.4. The Accuracy Verification

In order to verify the accuracy of the developed mathematical model of the BIW structure, the results of bending and torsion stiffness obtained by TBTMM are compared with the results obtained by FEM. The FEM analysis results are obtained by using the program Hypermesh, while the TBTMM analysis results are obtained through the MATLAB code. As listed in Table 1, it is reasonable to establish the vehicle BIW mathematical model by using TBTMM, as the relative error values are much lower than 20% [8].

**Table 1.** Material property of tread rubber.

| Stiffness Type | FEM Analysis | TBTMM Analysis | Error |
|---|---|---|---|
| Bending stiffness (N/mm) | 6515.3846 | 5963.5323 | −6.9% |
| Torsion stiffness (N.m/°) | 2642.9129 | 2697.2338 | 2.1% |

## 3. The Development of BHGA

GA is a traditional evolutionary algorithm, which is a stochastic search technique based on a series of possible solutions. Proposed by Holland in 1975 [9], GA has been extensively used in engineering and industry problems related to linear inequality constraints, nonlinear inequality constraints, equality constraints or unconstraints. Thus, it is reasonable to solve the constrained nonlinear optimization problem in equation (9) by using GA. As an unconstrained search technique, constrained problems have traditionally been challenging problems for GA. Several techniques of constraint handling have been developed: special representation and operator methods, penalty methods, separation of objective and constraint methods, repair methods, and hybrid methods. The most common way to introduce constraints in genetic algorithms is the penalty method, which punishes infeasible solutions by reducing their fitness values. Ersavas et al. [10] and Paszkowicz [11] use the static penalty method and dynamic penalty method for constraint optimization, respectively. However, as it is difficult to set the penalty coefficient properly, its performance is not always satisfactory. Considering this obstacle, Lin [12] came up with a rough penalty GA for constraint optimization; nevertheless, it requires another set of parameters to tune the penalty coefficients automatically. The main idea of the special representation and operator methods is to develop special representation schemes to tackle a certain problem for which generic representation schemes might not be appropriate. Koziel and Michalewicz [13] proposed a 'homomorphous map', in which they transformed the whole feasible region into a different shape that was easy to optimize. However, the implementation of the algorithm is more complex, and the experiments reported require a large number of fitness function evaluations. Although it is efficient for some intended applications, it can sometimes be difficult or even impossible to develop a special representation. The separation of objective and constraints methods treats the constraints as an objective function so that the original single-objective constraint optimization problem becomes a multi-objective unconstrained optimization problem, to which we can apply any multi-objective optimization techniques. Zhou et al. [14] developed a ranking procedure in accordance with the Pareto strength concept for the bi-objective problem, but with the constraints increasing, the objective function becomes complicated. As pointed out by Runarsson and Yao [15], the multi-objective techniques are difficult to find feasible solutions since most of the time is spent searching infeasible regions. While GA is coupled with another technique (e.g., another heuristic or a mathematical programming approach) to form hybrid methods, the new methods generally require several parameters to work properly, just like penalty methods. The main idea of the repair methods is to transform an infeasible solution into a feasible one, which can reduce the search space by using a repair technique. Furthermore, no special operators or modifications of the fitness function need to be considered in this case. Salcedo et al. [16] proposed a concept of a hybrid genetic algorithm in which the local search (LS) procedure is used as a constraint-handling technique. Later, Li et al. [17] proposed a boundary simulation genetic algorithm (BSGA) to address inequality constraints for GAs and developed a series of genetic operators that would abandon or repair infeasible individuals produced during the search process. However, it was not specified whether the infeasible solutions were abandoned or repaired, and it may not work properly for problems with disconnected feasible regions. Coello [18] has emphasized that a desired constraint-handling technique should be general and incorporate knowledge about the domain, efficiency, etc. Based on the BSGA and hybrid genetic algorithm, a boundary hybrid genetic algorithm (BHGA) that could be applied effectively to engineering is proposed in this paper. In general, the

method proposed in this paper has several features, which will be discussed in detail in the following chapters.

1.    The BHGA randomly selects individuals from the boundary point set as the feasible initial population and performs a global search (GS) using the GA;
2.    Perform the elitist strategy and adaptively tune crossover and mutation operators;
3.    The LS procedure is used to handle constraints.

### 3.1. Generate the Initial Population

In this paper, we mainly concentrate on the problems of inequality constraints. Thus, the mathematical form of the optimization problem could be formulated as the following minimum optimization problem:

Minimize: $f(\mathbf{x})$

Subject to:

$$\begin{cases} g_i(\mathbf{x}) \leq 0, & i = 1, 2, \ldots, m \\ x_j^l \leq x_j \leq x_j^u, & j = 1, 2, \ldots, n \end{cases} \tag{12}$$

where $\mathbf{x} = (x_1, x_2, \ldots x_n)$ is the vector of design variables; $m$ and $n$ are the number of constraints and design variables, respectively; $x_j^l$ and $x_j^u$ denote the lower and upper bounds of $x_j$, respectively.

#### 3.1.1. The Calculation of Boundary Points

Isaacs et al. [19] indicated that the optimal solutions to the constrained optimization problems are usually spread along the constraint boundary. The BSGA [18] is proposed to solve the constrained optimization problem based on the binary search method, but the binary search method would not search the maximum constraint boundary and may fall into the local optimum in the constraint boundary. Therefore, the reverse binary search method and LS strategy are developed in the BHGA, and it will be discussed in detail later. The flowchart for the main process of the generation of feasible regions is illustrated in Figure 5.

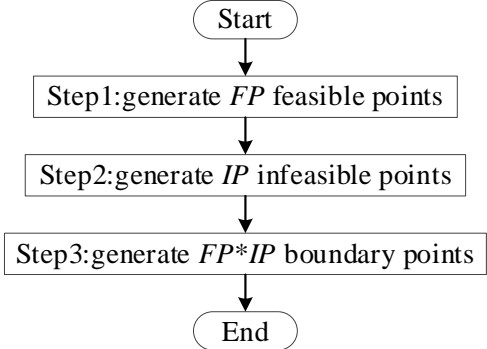

**Figure 5.** The flowchart for the calculation of boundary region.

The first step is to generate *FP* feasible points. Compared with unconstrained and simple constrained problems, it becomes relatively complicated to generate feasible points for complex constraints. In order to improve the search efficiency, the GA method is still used to generate the feasible points, and the main process is illustrated in Figure 6. Moreover, a real coding representation scheme is adopted in this paper. Initially, an empty set is developed for storing the feasible points and then generates the initial population, including *PS* individuals, randomly. The initial individuals satisfying the constraints are put into the feasible set, and the number of feasible points *NP* is recorded. The new populations are generated by selection, crossover, and mutation operators of GA [20], and the feasible points set is updated based on the feasible points in the new populations until the number of feasible points reaches *FP*.

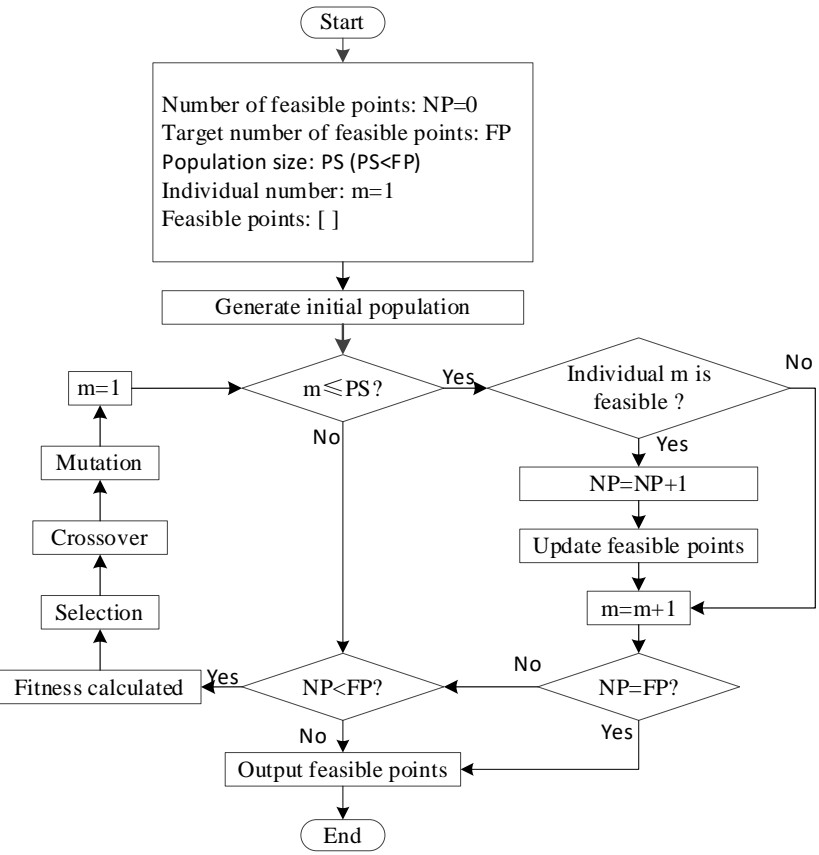

**Figure 6.** The flowchart for the generation of feasible points.

Since the GA is used to obtain a set of feasible points instead of an optimal solution, the fitness function needs to be modified. The aim is to reduce the differences between the individuals but not convergence, and the constraint violation status for each individual is expressed as Equation (13). The fitness value of every individual can be calculated by summing the constraint violation degree for each of them.

$$q_i(\mathbf{x}) = \begin{cases} g_i(\mathbf{x}) & \text{if } g_i(\mathbf{x}) > 0 \\ 0 & \text{if } g_i(\mathbf{x}) \le 0 \end{cases} \tag{13}$$

where $q_i(\mathbf{x})$ represents the $i$th constraint violation degree, while $g_i(\mathbf{x})$ represents the $i$th constraint.

The following step is to generate the infeasible points. Define the expansion region and transform the feasible region ($S_{feasible} = \{\mathbf{x} \in \mathrm{R}^n \,\big|\, x_j^l \le x_j \le x_j^u \text{ and } g_i(\mathbf{x}) \le 0, \text{ for } j = 1, 2, \ldots, n\}$) into infeasible region $\mathbf{S}'$:

$$\mathbf{S}' = \left\{ \mathbf{x} \in \mathrm{R}^n \,\big|\, \tilde{x}_j^l \le x_j \le \tilde{x}_j^u, \text{ for } j = 1, 2, \ldots, n \right\} \tag{14}$$

The $\tilde{x}_j^l$ and $\tilde{x}_j^u$ are the lower and upper extension region bounds of the $j$th design variable and are defined as Equation (15).

$$\begin{cases} \tilde{x}_j^l = x_j^{\min} - (x_j^u - x_j^l) \\ \tilde{x}_j^u = x_j^{\max} + (x_j^u - x_j^l) \end{cases} \tag{15}$$

where $x_j^{\min}$ and $x_j^{\max}$ are the minimum and maximum values of $j$th variable about the feasible point, respectively.

As shown in Figure 7, generate *IP* individuals in the extension region and set one of the variables in each individual to its lower or upper bound.

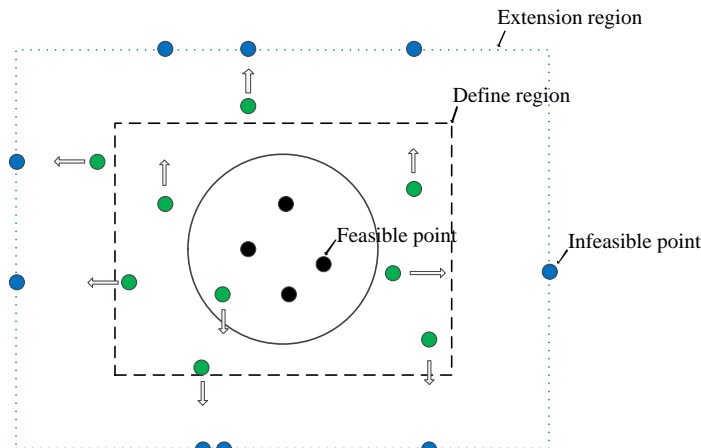

**Figure 7.** The distribution of point sets in the design domain.

Figure 8 illustrates the feasible point **a** moves to the constraint boundary by a large step size in the binary search method of BSGA, and the process ignores the possibility that there is a constraint boundary between the infeasible points **c** and **d**. Thus, in order to increase the search range of the feasible points (i.e., to find the maximum boundary of the feasible points) and reduce the step size, the reverse binary search method is proposed to find the boundary of the feasible region shown in Figure 8. Compared with the binary search method, the latter method requires $\varepsilon$ and $\Delta$ parameters to control the search accuracy. Moreover, the specific steps of the reverse binary search method are as follows:

Step 1: Choose a feasible point **a** and an infeasible point **b**, then go to step 2.

Step 2: Calculate the middle point **c** between **a** and **b**, then go to step 3.

Step 3: If **c** is feasible, **a** = **c**, then go to step 6; otherwise, calculate the middle point **d** between **c** and **b**, then go to step 4.

Step 4: If **d** is feasible, **a** = **d**, then go to step 6; otherwise, go to the next step.

Step 5: Calculate the distance between **b** and **d**; if it $\leq \Delta$, **b** = **d,** then go to step 2; otherwise, **c** = **d**, then go to step 3.

Step 6: Calculate the distance between **a** and **b**; if it $\leq \varepsilon$, terminate; otherwise, go to step 2.

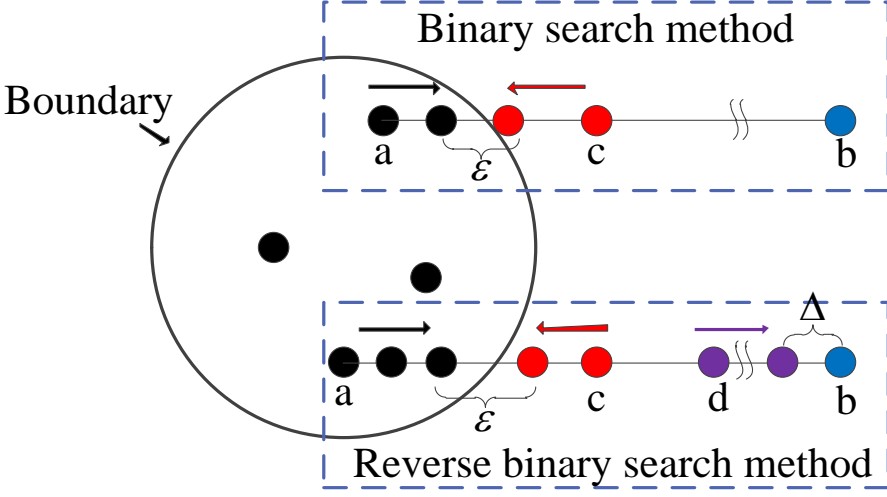

**Figure 8.** The difference between the binary search method and the reverse binary search method.

### 3.1.2. The Initial Population

A set of boundary points have been generated in the previous section, and then the initial population is randomly selected from the boundary points in the following steps. Initially, *popsize* individuals are randomly generated in the extension region, and the distances from each individual to all boundary points are calculated. Then the nearest boundary points are selected as the initial population individuals.

### 3.2. The Definition of GA Operators

The GA operators are mainly selection, crossover, and mutation operators. The selection operator is utilized to select some good individuals from the current population as parents to generate offspring. In this paper, the well-known roulette wheel selection operator is adopted [20]. Due to the randomness of the selection strategy, the elitist strategy is implemented by reinsertion to prevent the good individuals from being abandoned.

In the crossover and mutation operations, the setting of the crossover probability ($P_c$) and the mutation probability ($Pm$) is crucial to the generation of new individuals. In the BSGA method, the use of fixed crossover and mutation probabilities may lead to local optimum. The higher the value of $P_c$, the quicker the new individuals will be introduced into the population. However, as $P_c$ increases, individuals can be disrupted faster than selection can exploit them. Similarly, if the values of $P_m$ are too small, it is not easy to generate new individuals. However, the larger values of $P_m$ transform the GA into a pure random search algorithm. In order to improve the performance of the BHGA, the fixed values of $P_c$ and $P_m$ can no longer meet the dynamic performance of the algorithm, so the adaptive crossover and mutation probabilities are proposed in this paper.

$$P_c = \begin{cases} P_{c1} - \frac{(P_{c1}-P_{c2})(f'-f_{avg})}{f_{\max}-f_{avg}}, & f' \geq f_{avg} \\ P_{c1} & , \ f' < f_{avg} \end{cases} \tag{16}$$

$$P_m = \begin{cases} P_{m1} - \frac{(P_{m1}-P_{m2})(f'-f_{avg})}{f_{\max}-f_{avg}}, & f' \geq f_{avg} \\ P_{m1} & , \ f' < f_{avg} \end{cases} \tag{17}$$

where $f_{\max}$ is the maximum fitness value of the current population, while $f_{avg}$ is the average fitness value of the population and $f'$ is the individual fitness value; $P_{c1}$ and $P_{c2}$ represent the upper and lower limits of the crossover probability, while $P_{m1}$ and $P_{m2}$ represent the upper and lower limits of the mutation probability.

### 3.3. The LS Strategy

There are two main operators proposed to handle the infeasible individuals during the search process of BSGA: one is to regenerate it again until a feasible individual is obtained, but this will change the characteristics of the original individual; the other is to repair the infeasible individuals with the binary search method, but this will also cause the algorithm to converge to the boundary local optimum.

Thus, the LS procedure occurs as a method of constraint handling in the process of infeasible offspring approaching the feasible boundary in the BHGA. The current parent $P_r$ and the infeasible individuals $O_r$ generated by the crossover or mutation operation are selected to perform a nonlinear search along the $\overrightarrow{P_rO_r}$ direction and are defined as Equation (18). As shown in Figure 9, the dashed blue box represents the process of feasible individual $P_r$ searching for the boundary, while the red dashed box is the process of searching for the discrete feasible domain by the reverse binary search method. The main LS procedure is as follows:

Step 1: Select the current parent $P_r$ and the infeasible individual $O_r$, then go to step 2.

Step 2: Generate the repaired individual $P_o$ by Equation (18), and operate as follows

- If the $P_o$ is feasible, calculate the objective function value of $P_o$ and the distance between $P_o$ and $O_r$. If the current objective function value of $P_o$ is optimal compared

to other individuals or the distance $\leq$ the set value $\Delta$, then $P_o$ is output as the repaired individual; otherwise, $P_r = P_o$, and then go to step 2.

- If the $P_o$ is infeasible, then repair the infeasible individual with the reverse binary search method and go to step 2.

$$P_O = P_r + cr(O_r - P_r) \tag{18}$$

where $c$ is used to tune the LS step and $r$ is the random number which is uniformly distributed in the interval $[0, 1]$.

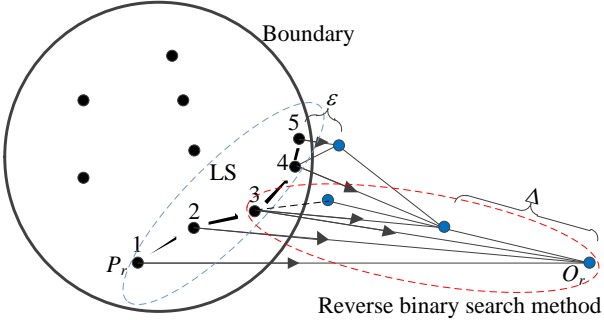

**Figure 9.** The process of the Local search.

## 4. Experimental Study and Discussion

In order to verify the numerical efficiency of the BHGA, 20 benchmark functions (6 unconstrained, 11 constrained, and 3 engineering-constrained problems, all of which are minimization problems) are adopted. The algorithm is running under Windows 7 Ultimate, and the code is programmed and compiled in MATLAB R2014a. All the details of the benchmark functions are listed in Table 2, where $N$ is the number of design variables; $LI$ and $NI$ are the number of linear inequality and nonlinear inequality constraints, respectively; $\rho$ is the estimated ratio between the feasible region and the search space; $BO$ represents the ratio between the number of constraints at the boundary with the total number of constraints. It can be seen from Table 2 that the optimal solution is easily located at the constraint boundary, so it is meaningful to propose the BHGA and start searching from the boundary point. All the used corresponding parameters of the BHGA are listed in Table 3.

**Table 2.** Details of benchmark functions.

| Prob. | N | Type of Function | LI | NI | ρ (%) | BO |
|-------|---|------------------|----|----|-------|-----|
| CF1 | 10 | nonlinear | 0 | 0 | 100 | - |
| CF2 | 10 | nonlinear | 0 | 0 | 100 | - |
| CF3 | 10 | nonlinear | 0 | 0 | 100 | - |
| CF4 | 10 | nonlinear | 0 | 0 | 100 | - |
| CF5 | 10 | nonlinear | 0 | 0 | 100 | - |
| CF6 | 10 | nonlinear | 0 | 0 | 100 | - |
| G01 | 13 | quadratic | 9 | 0 | 0.0111 | 6/9 |
| G02 | 20 | nonlinear | 0 | 2 | 99.9971 | 1/2 |
| G04 | 5 | quadratic | 0 | 6 | 52.1230 | 2/6 |
| G06 | 2 | cubic | 0 | 2 | 0.0066 | 2/2 |
| G07 | 10 | quadratic | 3 | 5 | 0.0003 | 6/8 |
| G08 | 2 | nonlinear | 0 | 2 | 0.8560 | 0/2 |
| G09 | 7 | polynomial | 0 | 4 | 0.5121 | 2/4 |
| G10 | 8 | linear | 3 | 3 | 0.0010 | 6/6 |
| G12 | 3 | quadratic | 0 | 1 | 4.7713 | 0/1 |
| G19 | 15 | nonlinear | 0 | 5 | 33.4761 | 5/5 |
| G24 | 2 | linear | 0 | 2 | 79.6556 | 2/2 |
| Eg01 | 3 | nonlinear | 1 | 3 | 0.7514 | 2/4 |
| Eg02 | 4 | cubic | 2 | 5 | 2.6627 | 4/7 |
| Eg03 | 4 | cubic | 3 | 1 | 75.9150 | 2/4 |

**Table 3.** The configurations of optimization.

| Prob. | FP | $P_{c1}$ | $P_{c2}$ | $P_{m1}$ | $P_{m2}$ | c | ε | Δ | Popsize | MaxGen |
|-------|-----|------|------|-----|------|---|--------|-----|---------|--------|
| CF1 | 40 | 0.95 | 0.85 | 0.2 | 0.01 | 1 | 0.1 | 1 | 30 | 1000 |
| CF2 | 40 | 0.95 | 0.85 | 0.2 | 0.01 | 1 | 0.1 | 1 | 30 | 1000 |
| CF3 | 40 | 0.95 | 0.85 | 0.2 | 0.01 | 1 | 0.1 | 1 | 30 | 1000 |
| CF4 | 40 | 0.95 | 0.85 | 0.2 | 0.01 | 1 | 0.1 | 1 | 30 | 1000 |
| CF5 | 40 | 0.95 | 0.85 | 0.2 | 0.01 | 1 | 0.1 | 1 | 30 | 1000 |
| CF6 | 40 | 0.95 | 0.85 | 0.2 | 0.01 | 1 | 0.1 | 1 | 30 | 1000 |
| G01 | 40 | 0.95 | 0.85 | 0.2 | 0.01 | 1 | 0.0001 | 0.1 | 200 | 1200 |
| G02 | 40 | 0.95 | 0.85 | 0.2 | 0.01 | 1 | 0.0001 | 0.1 | 200 | 1200 |
| G04 | 40 | 0.95 | 0.85 | 0.2 | 0.01 | 1 | 0.0001 | 0.1 | 200 | 1200 |
| G06 | 40 | 0.95 | 0.85 | 0.2 | 0.01 | 1 | 0.0001 | 0.1 | 200 | 1200 |
| G07 | 40 | 0.95 | 0.85 | 0.2 | 0.01 | 1 | 0.0001 | 0.1 | 200 | 1200 |
| G08 | 40 | 0.95 | 0.85 | 0.2 | 0.01 | 1 | 0.0001 | 0.1 | 200 | 1200 |
| G09 | 40 | 0.95 | 0.85 | 0.2 | 0.01 | 1 | 0.0001 | 0.1 | 200 | 1200 |
| G10 | 40 | 0.95 | 0.85 | 0.2 | 0.01 | 1 | 0.0001 | 0.1 | 200 | 1200 |
| G12 | 40 | 0.95 | 0.85 | 0.2 | 0.01 | 1 | 0.0001 | 0.1 | 200 | 1200 |
| G19 | 40 | 0.95 | 0.85 | 0.2 | 0.01 | 1 | 0.0001 | 0.1 | 200 | 1200 |
| G24 | 40 | 0.95 | 0.85 | 0.2 | 0.01 | 1 | 0.0001 | 0.1 | 200 | 1200 |
| Eg01 | 40 | 0.95 | 0.85 | 0.2 | 0.01 | 1 | 0.0001 | 0.1 | 100 | 200 |
| Eg02 | 40 | 0.95 | 0.85 | 0.2 | 0.01 | 1 | 0.0001 | 0.1 | 100 | 200 |
| Eg03 | 40 | 0.95 | 0.85 | 0.2 | 0.01 | 1 | 0.0001 | 0.1 | 100 | 200 |

*4.1. Benchmark of Unconstrained Functions*

For the unconstrained problem, six composite benchmark functions with several randomly located global and deep local optima of CEC'2005 [21] are considered in this paper. Considering these unconstrained optimization problems is mainly to verify whether the BHGA is more advantageous than the BSGA for optimal local problems and whether it can be used to deal with unconstrained optimization problems.

The statistic of optimization results of the BHGA and the comparison algorithms are plotted in Table 4. 10 comparison algorithms are adopted, which are BSGA [17], IGA [22], SSA [23], GOA [24], WOA [25], GWO [26], PSO [27], GSA [28], MVO [29], and HS [30]. For all the algorithms, the same population size and iteration number equal to 30 and 1000 have been utilized, and all algorithms run 30 times. The optimization results of the comparison algorithms use the results of the literature [22].

**Table 4.** Statistical results of composite benchmark functions.

| Algorithm | Statistic | F1 | F2 | F3 | F4 | F5 | F6 |
|-----------|-----------|------|------|------|------|------|------|
| BHGA | Ave. | 43.67 | 96.27 | 198.42 | 417.13 | 64.99 | 535.21 |
| | Std. | 68.20 | 84.20 | 58.13 | 78.65 | 75.18 | 91.25 |
| | Rank | 3 | 3 | 4 | 6 | 2 | 1 |
| BSGA | Ave. | 66.67 | 123.75 | 205.02 | 572.80 | 410.51 | 819.30 |
| | Std. | 118.42 | 150.46 | 75.75 | 140.74 | 83.57 | 162.49 |
| | Rank | 5 | 4 | 6 | 10 | 11 | 8 |
| IGA | Ave. | 46.67 | 89.10 | 191.15 | 344.55 | 90.21 | 544.26 |
| | Std. | 77.61 | 4.10 | 61.34 | 54.47 | 57.41 | 68.40 |
| | Rank | 4 | 2 | 2 | 2 | 4 | 2 |
| SSA | Ave. | 36.67 | 303.35 | 240.79 | 334.99 | 28.18 | 608.64 |
| | Std. | 55.56 | 373.25 | 86.51 | 30.45 | 33.28 | 18.42 |
| | Rank | 2 | 11 | 7 | 1 | 1 | 3 |
| GOA | Ave. | 120.00 | 261.49 | 341.83 | 516.50 | 195.43 | 846.43 |
| | Std. | 121.48 | 121.63 | 132.22 | 166.93 | 191.41 | 13.79 |
| | Rank | 10 | 10 | 10 | 8 | 9 | 9 |
| WOA | Ave. | 120.92 | 173.93 | 417.50 | 610.51 | 141.62 | 673.17 |
| | Std. | 129.64 | 91.30 | 157.50 | 138.71 | 122.15 | 196.52 |
| | Rank | 11 | 7 | 11 | 11 | 7 | 4 |

**Table 4.** *Cont.*

| Algorithm | Statistic | F1 | F2 | F3 | F4 | F5 | F6 |
|---|---|---|---|---|---|---|---|
| GWO | Ave. | 82.19 | 146.49 | 200.72 | 430.56 | 93.48 | 860.55 |
| | Std. | 115.17 | 93.06 | 72.71 | 125.21 | 102.17 | 123.24 |
| | Rank | 6 | 6 | 5 | 7 | 5 | 11 |
| PSO | Ave. | 113.33 | 124.15 | 191.89 | 346.67 | 131.98 | 751.69 |
| | Std. | 107.42 | 96.56 | 79.37 | 102.83 | 106.71 | 194.32 |
| | Rank | 9 | 5 | 3 | 3 | 6 | 5 |
| GSA | Ave. | 3.33 | 186.67 | 157.14 | 410.00 | 195.43 | 814.97 |
| | Std. | 18.26 | 50.74 | 55.12 | 156.06 | 191.41 | 113.47 |
| | Rank | 1 | 9 | 1 | 5 | 10 | 6 |
| MVO | Ave. | 86.68 | 177.41 | 299.11 | 392.93 | 86.15 | 815.65 |
| | Std. | 81.72 | 118.18 | 154.94 | 126.35 | 112.38 | 16.17 |
| | Rank | 7 | 8 | 9 | 4 | 3 | 7 |
| HS | Ave. | 93.25 | 75.29 | 273.79 | 524.60 | 193.47 | 846.86 |
| | Std. | 44.60 | 255.75 | 101.48 | 138.01 | 128.06 | 12.77 |
| | Rank | 8 | 1 | 8 | 9 | 8 | 10 |

From the ranking of the average results in Table 4, it can be seen that although the BHGA has not obtained an optimal solution, the stability of the solution for different unconstrained problems is generally acceptable. Moreover, the BSGA is more likely to fall into the optimal local solution for composite functions, and the proposed BHGA is more effective.

*4.2. Benchmark of Constrained Functions*

The 11 constrained problems (G01, G02, G04, G06-G10, G12, G19, G24) selected from the CEC'2006 [31] are adopted to verify the constrained optimization ability of the BHGA. The optimal results (best, worst, average, and standard deviation) obtained by the BHGA are listed in Table 5, and the algorithm runs 30 times with 240,000 function evaluations. The exact/near-optimal results calculated by the BHGA are highlighted in boldface, and 8 of the 11 benchmark functions (G01, G02, G04, G06, G08, G09, G12, G24) obtained the known optimal results. Only G10 did not obtain the exact/near-optimal result, but it is also hard for G10 to get the optimal result through other algorithms. Table 6 shows the comparative results of benchmark functions obtained by the BHGA and other comparison algorithms (d-DS [32], BSGA [17], HTS [33], BBO [33], TLBO [33], GA [34], PSO [35], DE [36], and ABC [37]). As an algorithm that mainly solves constraint problems, the BHGA has a great improvement in computing ability compared with GA, and it has better computing stability than the BSGA.

**Table 5.** Results obtained by the BHGA algorithm for 11 benchmark functions over 30 independent runs with 240,000 function evaluations.

| Prob. | Opti. | Best | Worst | Mean | SD |
|---|---|---|---|---|---|
| G01 | −15 | **15.0000** | −15.0000 | −15.0000 | $1.9 \times 10^{-6}$ |
| G02 | −0.8036 | **−0.8036** | −0.7926 | −0.8008 | $4.5 \times 10^{-3}$ |
| G04 | −30,665.5387 | **−30,665.5387** | −30,665.5387 | −30,665.5387 | $4.5 \times 10^{-6}$ |
| G06 | −6961.8139 | **−6961.8139** | −6961.8139 | −6961.8139 | $4.5 \times 10^{-6}$ |
| G07 | 24.3062 | 24.3078 | 24.9419 | 24.4864 | $1.9 \times 10^{-1}$ |
| G08 | −0.095825 | **−0.095825** | −0.095825 | −0.095825 | $3.1 \times 10^{-11}$ |
| G09 | 680.6301 | **680.6301** | 680.6626 | 680.6408 | $8.1 \times 10^{-3}$ |
| G10 | 7049.2480 | 7114.8305 | 7795.3801 | 7354.5570 | $1.2 \times 10^{-2}$ |
| G12 | −1 | **−1.0000** | −1.0000 | −1.0000 | $6.1 \times 10^{-13}$ |
| G19 | 32.6556 | 32.8912 | 38.8550 | 35.3045 | 1.5 |
| G24 | −5.5080 | **−5.5080** | −5.5080 | −5.5080 | $3.7 \times 10^{-8}$ |

**Table 6.** Comparative results of benchmark functions obtained by different algorithms.

| Prob. | | BHGA | d-DS | BSGA | HTS | GA | PSO | DE | ABC | BBO | TLBO |
|---|---|---|---|---|---|---|---|---|---|---|---|
| g01 | Best | **−15.0000** | **−15** | −14.9999 | **−15** | 14.44 | **−15** | **−15** | **−15** | −14.977 | **−15** |
| | Worst | −15.0000 | −13 | −14.9996 | −15 | | −13 | −11.828 | −15 | −14.5882 | −6 |
| | Mean | −15.0000 | −12.3 | −14.9997 | −15 | −14.236 | −14.71 | −14.555 | −15 | −14.7698 | −10.782 |
| | SD | $1.9 \times 10^{-6}$ | $1.8 \times 10^{-2}$ | $6.7 \times 10^{-5}$ | - | | - | - | - | - | - |
| g02 | Best | **−0.8036** | −0.803518 | **−0.8036** | −0.7515 | −0.796321 | −0.669158 | −0.472 | **−0.803598** | −0.7821 | −0.7835 |
| | Worst | −0.7926 | −0.7743 | −0.7215 | −0.5482 | − | −0.299426 | - | −0.749797 | −0.7389 | −0.5518 |
| | Mean | −0.8008 | −0.7880 | −0.7669 | −0.6437 | −0.788588 | −0.41996 | −0.655 | −0.792412 | −0.7642 | −0.6705 |
| | SD | $4.5 \times 10^{-3}$ | $7.0 \times 10^{-4}$ | $2.3 \times 10^{-2}$ | - | - | - | - | - | - | - |
| g04 | Best | **−30,665.5387** | −30,665.539 | −30,665.5385 | **−30,665.5387** | −30626.053 | **−30,665.539** | **−30,665.539** | **−30,665.539** | **−30,665.539** | **−30,665.539** |
| | Worst | −30,665.5387 | −30,665.6475 | −30,665.5380 | −30,665.5387 | − | −30,665.539 | −30,665.539 | −30,665.539 | −29942.3 | −30,665.539 |
| | Mean | −30,665.5387 | −30,665.8862 | −30,665.5383 | −30,665.5387 | −30590.455 | −30,665.539 | −30,665.539 | −30,665.539 | −30411.865 | −30,665.539 |
| | SD | $4.5 \times 10^{-6}$ | $1.2 \times 10^{-1}$ | $1.3 \times 10^{-4}$ | - | - | - | - | - | - | - |
| G06 | Best | **−6961.8139** | **−6961.8139** | −6961.6025 | −6961.814 | −6952.472 | **−6961.814** | −6954.434 | **−6961.814** | **−6961.814** | **−6961.814** |
| | Worst | −6961.8139 | 3.6128E+07 | −6959.5077 | −6961.814 | − | −6961.814 | −6954.434 | −6961.814 | −5404.4941 | −6961.814 |
| | Mean | −6961.8139 | 1.8436E+06 | −6961.1706 | −6961.814 | −6872.204 | −6961.814 | −6954.434 | −6961.814 | −6181.7461 | −6961.814 |
| | SD | $4.5 \times 10^{-6}$ | $8.1 \times 10^{6}$ | $4.5 \times 10^{-1}$ | - | - | - | - | - | - | - |
| g07 | Best | 24.3078 | 24.315 | 24.3250 | 24.3104 | 31.097 | 24.37 | 24.306 | 24.33 | 25.6645 | 24.3103 |
| | Worst | 24.9419 | 25.5336 | 36.3810 | 25.0083 | - | 56.055 | 24.33 | 25.19 | 37.6912 | 27.6106 |
| | Mean | 24.4864 | 24.7153 | 25.3126 | 24.4945 | 34.98 | 32.407 | 24.31 | 24.473 | 29.829 | 24.837 |
| | SD | $1.9 \times 10^{-1}$ | $3.1 \times 10^{-2}$ | 2.2 | - | - | - | - | - | - | - |
| g08 | Best | **−0.095825** | **−0.095825** | **−0.095825** | **−0.095825** | **−0.095825** | **−0.095825** | **−0.095825** | **−0.095825** | **−0.095825** | **−0.095825** |
| | Worst | −0.095825 | −0.09582 | −0.095825 | −0.095825 | - | −0.095825 | −0.095825 | −0.095825 | −0.095817 | −0.095825 |
| | Mean | −0.095825 | −0.0958 | −0.095825 | −0.095825 | −0.095799 | −0.095825 | −0.095825 | −0.095825 | −0.095824 | −0.095825 |
| | SD | $3.1 \times 10^{-11}$ | 0 | $7.2 \times 10^{-11}$ | - | - | - | - | - | - | - |
| g09 | Best | **680.6301** | **680.630** | 680.6321 | **680.6301** | 685.994 | 680.63 | 680.63 | 680.634 | **680.6301** | **680.6301** |
| | Worst | 680.6626 | 681.1324 | 680.7393 | 680.644 | - | 680.631 | 680.631 | 680.653 | 721.0795 | 680.6456 |
| | Mean | 680.6408 | 680.7132 | 680.6587 | 680.6329 | 692.064 | 680.63 | 680.63 | 680.64 | 692.7162 | 680.6336 |
| | SD | $8.1 \times 10^{-3}$ | $1.1 \times 10^{-3}$ | $2.6 \times 10^{-2}$ | - | - | - | - | - | - | - |
| g10 | Best | 7114.8305 | 7056.76 | 7479.5547 | 7049.4836 | 9079.77 | 7049.481 | 7049.548 | 7053.904 | 7679.0681 | 7250.9704 |
| | Worst | 7795.3801 | 7846.7898 | 10074.6906 | 7252.0546 | - | 7894.812 | 9264.886 | 7604.132 | 9570.5714 | 7291.3779 |
| | Mean | 7354.5570 | 7350.3449 | 8945.5845 | 7119.7015 | 10003.225 | 7205.5 | 7147.334 | 7224.407 | 8764.9864 | 7257.0927 |
| | SD | $1.2 \times 10^{-2}$ | $2.0 \times 10^{1}$ | $8.0 \times 10^{2}$ | − | - | - | - | - | - | - |
| g12 | Best | **−1.0000** | **−1** | **−1** | **−1** | **−1** | **−1** | **−1** | **−1** | **−1** | **−1** |
| | Worst | −1.0000 | −1 | −1 | −1 | −1 | −0.994 | −1 | −1 | −1 | −1 |
| | Mean | −1.0000 | −1 | −1 | −1 | −1 | −0.998875 | −1 | −1 | −1 | −1 |
| | SD | $6.1 \times 10^{-13}$ | 0 | $6.0 \times 10^{-13}$ | - | - | - | - | - | - | - |
| g19 | Best | 32.8912 | **32.6556** | 33.5364 | 32.7132 | - | 33.5358 | 32.6851 | 33.3325 | 39.1471 | 32.7916 |
| | Worst | 38.8550 | 46.1658 | 47.2062 | 33.2140 | - | 39.8443 | 32.9078 | 38.5614 | 71.3106 | 36.1935 |
| | Mean | 35.3045 | 32.8047 | 37.4585 | 32.7903 | - | 36.6172 | 32.7680 | 36.0078 | 51.8769 | 34.0792 |
| | SD | 1.5 | 2.8 | 3.3 | - | - | 2.04 | $6.28 \times 10^{-2}$ | 1.83 | $1.12 \times 10^{1}$ | $9.33 \times 10^{-1}$ |
| g24 | Best | **−5.5080** | **−5.5080** | **−5.5080** | **−5.5080** | - | **−5.5080** | **−5.5080** | **−5.5080** | **−5.5080** | **−5.5080** |
| | Worst | −5.5080 | −5.4661 | −5.5080 | −5.5080 | - | −5.5080 | −5.5080 | −5.5080 | −5.4857 | −5.5080 |
| | Mean | −5.5080 | −5.5080 | −5.5080 | −5.5080 | - | −5.5080 | −5.5080 | −5.5080 | −5.4982 | −5.5080 |
| | SD | $3.7 \times 10^{-8}$ | $3.4 \times 10^{-6}$ | $3.5 \times 10^{-6}$ | - | - | $9.36 \times 10^{-16}$ | $9.36 \times 10^{-16}$ | $9.36 \times 10^{-16}$ | $6.75 \times 10^{-3}$ | $9.36 \times 10^{-16}$ |

### 4.3. Benchmark of Constrained Engineering Functions

In this section, the BHGA was also tested with three constrained engineering problems [26]: a tension/compression spring (Eg01), a welded beam (Eg02), and a pressure vessel (Eg03).

The comparison of the BHGA optimization results with literature for the engineering problem is listed in Tables 7–9, where the bold number indicates the best results. Inspecting the results of the algorithms on those problems makes it evident that the BHGA managed to show very competitive results compared to IGA [22], BSGA [17], GA [38], GA [18], and TLBO [39], and obtains a better result on the pressure vessel problem. As an improved GA, the BHGA obtains better results than GA with fewer evaluations and also has a great improvement in computing ability compared with the BSGA. Taken together, the BHGA is efficient as a constrained handling method, especially for engineering constraint problems and constrained problems.

**Table 7.** Comparison of the BHGA optimization results with literature for the tension/compression spring problem.

| Algorithm | Design Variables | | | Optimum Result | Max. Eval. |
|---|---|---|---|---|---|
| | $x_1$ | $x_2$ | $x_3$ | | |
| BHGA | 0.051702 | 0.357034 | 11.270455 | **0.012665** | 20,000 |
| IGA | 0.051760 | 0.358421 | 11.191034 | 0.012667 | 50,000 |
| BSGA | 0.052499 | 0.376505 | 10.216692 | 0.012677 | 20,000 |
| GA (2000) | - | - | - | 0.012822 | 900,000 |
| GA (2002) | 0.051989 | 0.363965 | 10.890522 | 0.012973 | 80,000 |
| TLBO | - | - | - | **0.012665** | 10,000 |

**Table 8.** Comparison of the BHGA optimization results with literature for the welded beam problem.

| Algorithm | Design Variables | | | | Optimum Result | Max. Eval. |
|---|---|---|---|---|---|---|
| | $x_1$ | $x_2$ | $x_3$ | $x_4$ | | |
| BHGA | 0.205711 | 3.470841 | 9.036781 | 0.205729 | 1.724893 | 20,000 |
| IGA | 0.205218 | 3.481537 | 9.036823 | 0.205731 | 1.725597 | 50,000 |
| BSGA | 0.191842 | 3.802379 | 9.023441 | 0.206332 | 1.749193 | 20,000 |
| GA (2000) | - | - | - | - | 1.748309 | 900,000 |
| GA (2002) | 0.205986 | 3.471328 | 9.020224 | 0.206480 | 1.728226 | 80,000 |
| TLBO | - | - | - | - | **1.724852** | 10,000 |

**Table 9.** Comparison of the BHGA optimization results with literature for the pressure vessel problem.

| Algorithm | Design Variables | | | | Optimum Result | Max. Eval. |
|---|---|---|---|---|---|---|
| | $x_1$ | $x_2$ | $x_3$ | $x_4$ | | |
| BHGA | 0.789938 | 0.390530 | 40.9293 | 191.683131 | **5905.9633** | 20,000 |
| IGA | 0.815752 | 0.403932 | 42.248583 | 174.814712 | 5957.9898 | 50,000 |
| BSGA | 0.8074 | 0.3990 | 41.8153 | 180.1774 | 5939.1857 | 20,000 |
| GA (2000) | 0.812500 | 0.434500 | 40.323900 | 200.000000 | 6288.7445 | 900,000 |
| GA (2002) | 0.812500 | 0.437500 | 42.097398 | 176.654050 | 6059.9463 | 80,000 |
| TLBO | - | - | - | - | 6059.714335 | 10,000 |

## 5. Lightweight Design Based on BHGA

In this chapter, the BHGA method is adopted to lighten the BIW mass. As the symmetry of the BIW structure, the design variables of beam members (1, 2, 3, 4, 5, 8, 9, 10, 48) are defined to have the same properties as beam members (17, 18, 19, 20, 21, 24, 25, 26, 49). Take the bending stiffness and the torsion stiffness as the constraints, and set the allowable limit values of constraint condition with respect to bending stiffness and torsion stiffness are 6000 N/mm and 2500 Nm/deg, respectively. The configurations of the BHGA are listed in Table 10.

**Table 10.** The configurations of BIW optimization.

| Prob. | FP | $P_{c1}$ | $P_{c2}$ | $P_{m1}$ | $P_{m2}$ | c | ε | Δ | Popsize | MaxGen |
|---|---|---|---|---|---|---|---|---|---|---|
| BIW | 20 | 0.95 | 0.85 | 0.2 | 0.01 | 1 | 0.0001 | 0.5 | 40 | 200 |

The mass convergence curve for the optimization process is obtained, as depicted in Figure 10. Moreover, the optimized values of the cross-sections are listed in Table 11, where the initial and bounds values of design variables are also listed. According to the optimized and initial values of the cross-sections, it can be seen that the mass of the auto-body decreases by about 14.8 kg (from 111.6 kg to 96.8 kg). Consequently, the results indicate that it is effective to use the BHGA for the lightweight design of the BIW structure.

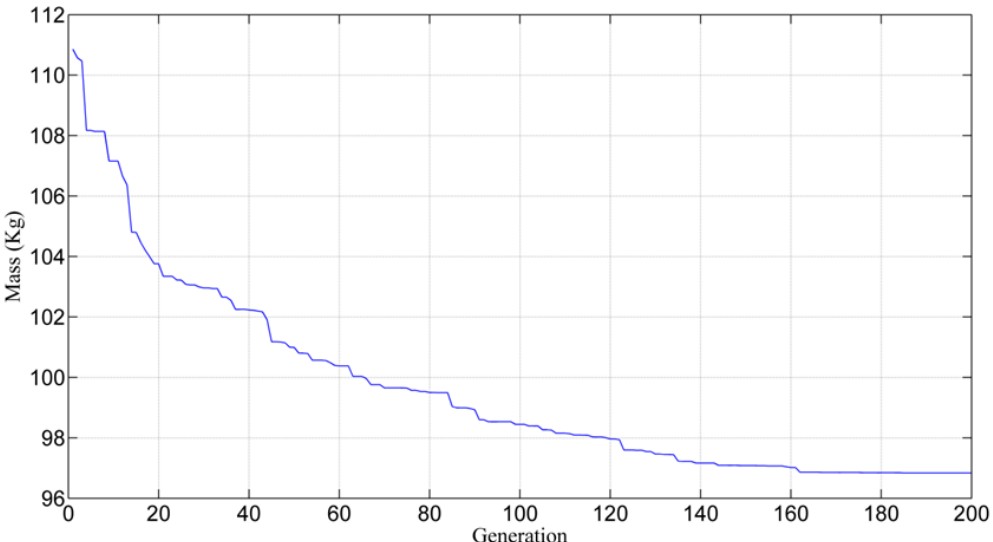

**Figure 10.** The convergence of object function.

**Table 11.** The initial bounds and optimized values of the design variables.

| No. | Design Variable | | | | | | | | | | | |
|---|---|---|---|---|---|---|---|---|---|---|---|---|
| | a (mm) | | | | b (mm) | | | | t (mm) | | | |
| | Initial | LB | UB | Optimum | Initial | LB | UB | Optimum | Initial | LB | UB | Optimum |
| 1 | 80 | 70 | 90 | 70.0091 | 50 | 40 | 60 | 40.3296 | 2 | 1 | 3 | 1.2210 |
| 2 | 50 | 40 | 60 | 40.0002 | 60 | 50 | 70 | 50.0298 | 2 | 1 | 3 | 1.1240 |
| 3 | 50 | 40 | 60 | 58.9283 | 60 | 50 | 70 | 50.3071 | 2 | 1 | 3 | 1.5336 |
| 4 | 50 | 40 | 60 | 43.0401 | 100 | 90 | 110 | 90.0902 | 2 | 1 | 3 | 1.4500 |
| 5 | 50 | 40 | 60 | 40.0000 | 140 | 130 | 150 | 130.0087 | 2 | 1 | 3 | 1.1200 |
| 6 | 50 | 40 | 60 | 59.0542 | 80 | 70 | 90 | 70.3845 | 2 | 1 | 3 | 1.6700 |
| 7 | 50 | 40 | 60 | 50.4979 | 80 | 70 | 90 | 77.0980 | 2 | 1 | 3 | 1.7085 |
| 8 | 50 | 40 | 60 | 40.0013 | 80 | 70 | 90 | 70.7770 | 2 | 1 | 3 | 2.2068 |
| 9 | 50 | 40 | 60 | 50.8359 | 60 | 50 | 70 | 50.0000 | 2 | 1 | 3 | 1.1000 |

## 6. Conclusions

The TBTMM is used to establish the relationship between the section properties and structural mechanical performances to improve the calculation accuracy. The bending and torsional stiffness errors of the mathematical simulation model and the finite element model are 6.9% and 2.1%, respectively, which are within the reasonable error range. Moreover, as the optimal solutions of the constrained optimization problems are usually distributed along the constraint boundary, a more general and simpler constrained optimization algorithm BHGA is proposed based on a hybrid genetic algorithm and LS. Twenty problems (six unconstrained, eleven constrained, and three engineering-constrained problems) are benchmark tested and compared with well-regarded algorithms, which proves the effectiveness of this algorithm. Finally, BHGA is used in the lightweight design of vehicle BIW, providing initial design parameters for the vehicle conceptual design stage.

This paper only considers the static performances of BIW with single-cell thin-walled beams and describes its lightweight design as a single objective optimization problem. In further studies, we will focus on two aspects. On the one hand, we will improve the mechanical model of the BIW, such as establishing the static transfer matrix and dynamic transfer matrix of thin-walled beams with arbitrary cross-sections, improving the joint mechanical transfer model of the thin-walled beam, and establishing the transfer matrix of the curved thin-walled beam. On the other hand, we will combine other genetic operators and diversity-maintaining strategies to improve the BHGA and develop its parallel processing capability.

**Author Contributions:** Conceptualization, T.X. and M.S.; methodology, H.Z. and M.S.; software, H.Z.; validation, T.X.; M.S., H.Z. and J.Y.; formal analysis, H.Z. and J.Y.; investigation, H.Z.; resources, T.X.; data curation, H.Z.; writing—original draft preparation, H.Z.; writing—review and editing, T.X. and F.G.; visualization, M.S.; supervision, T.X.; project administration, T.X.; funding acquisition, T.X. All authors have read and agreed to the published version of the manuscript.

**Funding:** This research was funded by "Research and development of energy-saving and environment-friendly high-performance non-pneumatic tire" and "Research and development of a new type of non pneumatic tire applied to micro vehicles". The grant numbers are X220091TL220 and X201011XQ200 respectively.

**Data Availability Statement:** Data are available upon request from the authors.

**Conflicts of Interest:** The authors declare no conflict of interest.

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
