# Peer review of "Optimization Design of Automotive Body Stiffness Using a Boundary Hybrid Genetic Algorithm"

_machines, doi:10.3390/machines10121171_

Round 1
Reviewer 1 Report
In this paper the Authors, based on the thin-walled beam transfer matrix method, establish the relationship between the section properties and structural mechanical performances in order to improve the calculation accuracy of vehicle structure.
The manuscript is interesting, fits well with the aim of the “Machines" Journal, and it is opinion of the reviewer that it can be published after the following minor revisions.
(1) In Figure 1, the drawing of the cross-section could be placed sideways to the model of the structure, so as not to have overlapping.
(2) In Equation 7, the matrix dimension should be placed below the matrix. The Authors should better explain why it has dimension 1484x742, when the T matrix has index from 1 to 53.
(3) Table 1 should also indicate the units of measurement.
(4) The flow chart in Figure 6 should be graphically revised. After the first IF (m≤PS?) it is better to continue vertically with "NO" and have the lateral control loop. This improves the readability.
(5) It is recommended to partition Table 3 with horizontal thin lines to separate the data for each type of algorithm. It is also recommended not to use scientific notation and to always use the same number of decimal places. This improves the readability of the data.
(6) It is recommended to partition Table 5 with horizontal thin lines to separate the data for each benchmark functions.
(7) The table in Appendix A could be included in the text of the manuscript (removing the appendix section).
According to what said above, the reviewer’s opinion is that the manuscript can be accepted for publication after the described minor revisions.
Author Response
Firstly, thank you very much for your comments concerning our manuscript. Those comments are all valuable and very helpful for revising and improving our paper, as well as the important guiding significance to our researches.
The response to the reviewer's comments has been listed in the attachment. Please see the attachment.

Reviewer 2 Report
- To what extent does the use of the genetic algorithm technique present the most significant challenges in the proposed BSGA?
- Ensure that the most recent reference is included in the article.
- In what ways are hybrid genetic algorithms useful in the simplified modular BIW conceptual model?
- Where do these findings fit in the larger scheme of things? How, in other words, might one put these findings into practice? Case studies addressing the aforementioned needs are to be provided by the author.
- It is suggested that the paper be proofread by someone who knows a lot about English in order to fix the grammar mistakes.
- In light of the current limitations of the research, please discuss the long-term objectives.
- Prior research findings are discussed to the extent necessary for readers to comprehend the current study's justification and methodology. The authors make a systematic addition to the research that has already been done in this field.
Author Response

(The authors gave the same response as above.)

Reviewer 3 Report
The paper investigated a thin-walled beam transfer matrix method and its adaptation to set out a vehicle body-in-white structure’s mathematical model. This simplified modular conceptual model has been compared with the results of finite element method for verifying of approach.
The authors have referred 39 publications, which are enough and advisable.
The paper is understandable, well structured and sufficiently detailed.
The theoretical and practical conclusions drawn are relevant and important.
My suggestions to improve the quality of paper:
1. Please, use section Acronyms. Several initial words are not interpreted (for example: rows 12; 362-363).
2. Figures 1; 3; 4; 6; 9; 10 should be enlarged for better understanding.
3. Rows 95 to 106 should be reformatted.
4. Please, rewrite “kth” in the row 93.
5. The sizes of the vectors and matrices in equations (6) - (9) should be made clear.
6. The equations in rows 217; 254-255 (between brackets); 308; 309 must be numbered.
7. I suggest, the relative errors can be enounced using absolute values (see row 431).
8. Please, write the word “equation” instead “Eq.” – it is a more elegant form.
9 In Figure 5., please, use “FP . IP” form.
10. Conclusions should be expanded by deductions in more details and suggestions for future works.

Author Response

(The authors gave the same response as above.)

Round 2
Reviewer 2 Report
Accept the revision copy. Also recommended for publication